# Promising Phytogenic Feed Additives Used as Anti-Mycotoxin Solutions in Animal Nutrition

**DOI:** 10.3390/toxins16100434

**Published:** 2024-10-10

**Authors:** Sergio Quesada-Vázquez, Raquel Codina Moreno, Antonella Della Badia, Oscar Castro, Insaf Riahi

**Affiliations:** Bionte Nutrition, 43204 Reus, Spain; serquesada90@gmail.com (S.Q.-V.); raquel.codina@bionte.com (R.C.M.); antonella.dellabadia@bionte.com (A.D.B.);

**Keywords:** phytogenics, bioactive compounds, mycotoxins, animals, livestock, aquaculture, pet, biosorbents

## Abstract

Mycotoxins are a major threat to animal and human health, as well as to the global feed supply chain. Among them, aflatoxins, fumonisins, zearalenone, T-2 toxins, deoxynivalenol, and Alternaria toxins are the most common mycotoxins found in animal feed, with genotoxic, cytotoxic, carcinogenic, and mutagenic effects that concern the animal industry. The chronic negative effects of mycotoxins on animal health and production and the negative economic impact on the livestock industry make it crucial to develop and implement solutions to mitigate mycotoxins. In this review, we summarize the current knowledge of the mycotoxicosis effect in livestock animals as a result of their contaminated diet. In addition, we discuss the potential of five promising phytogenics (curcumin, silymarin, grape pomace, olive pomace, and orange peel extracts) with demonstrated positive effects on animal performance and health, to present them as potential anti-mycotoxin solutions. We describe the composition and the main promising characteristics of these bioactive compounds that can exert beneficial effects on animal health and performance, and how these phytogenic feed additives can help to alleviate mycotoxins’ deleterious effects.

## 1. Introduction

Mycotoxins are toxic secondary metabolites produced by filamentous fungi, primarily from the genera *Aspergillus*, *Penicillium*, and *Fusarium* [1,2]. These low molecular weight compounds pose a serious threat to the health of both animals and humans. One of the most concerning aspects of mycotoxins in animal production is their physical and chemical properties, which make them highly stable during agricultural processes. This stability allows them to persist in cereal grains throughout harvesting, transportation, and storage, leading to long-term contamination [2]. As a result, mycotoxins can be transferred from feed to livestock and even into dairy products, posing a significant risk to the global feed supply chain. This affects not only feed safety but also animal health and productivity, ultimately impacting human health through animal-derived products, economies, and international trade [2].

The health consequences of mycotoxin exposure can be classified as either acute or chronic, with chronic toxicity posing a particular concern in livestock production. Chronic exposure to mycotoxins can negatively affect animal appetite and lead to a range of metabolic, physiological, and immunological disorders. These disruptions result in reduced production efficiency, including lower feed intake, poor growth rates, decreased egg production, changes in carcass quality, and reduced fertility, ultimately leading to economic losses [3].

The principal mycotoxins contaminating livestock feed include *Fusarium*-derived toxins such as fumonisins (FBs), trichothecenes [deoxynivalenol (DON) and T-2/HT-2 toxin)], and zearalenone (ZEN), as well as non-*Fusarium* mycotoxins like ochratoxins (OTs) and aflatoxins (AFs) [4]. They are strictly regulated in parts of the European Union and North America due to their impact on livestock feed contamination. Regulatory bodies overseeing food and agriculture have set maximum permissible limits for each type of mycotoxin in animal feed and food products to prevent mycotoxicosis in livestock [4,5]. In recent years, however, attention has shifted to emerging mycotoxins such as those produced by *Alternaria* species, particularly alternariol (AOH), alternariol monomethyl ether (AME), and tenuazonic acid (TeA). These toxins have become potential contaminants in livestock feed, prompting increased interest in their biomonitoring and regulation [6].

Environmental conditions play a key role in the growth of mycotoxin-producing fungi. Warm and humid climates, typically found in subtropical and tropical regions, create ideal conditions for fungal proliferation and mycotoxin production [7]. Furthermore, climate change and global warming are altering the climatic patterns of many regions, including those in temperate zones that traditionally experience cooler, drier climates. These changes have led to an increase in mycotoxicosis outbreaks in agricultural commodities and animal feed in these regions, making it crucial to continuously monitor climate trends and food contamination [8]. Controlling the risk of mycotoxin development and mitigating unexpected outbreaks due to climate change and globalization are vital for maintaining feed safety.

Several strategies have been developed to mitigate the detrimental effects of mycotoxins in animal feed and consequently protect animal and human health. These strategies can be broadly classified into pre-harvest and post-harvest measures. Pre-harvest strategies include the use of resistant crop varieties, crop rotation, and the application of fungicides to reduce fungal infection and mycotoxin production [9]. Post-harvest strategies involve proper drying and storage conditions to inhibit fungal growth, the use of chemical and biological agents to degrade or adsorb mycotoxins, and the implementation of rigorous monitoring and regulatory frameworks to ensure feed safety [10].

Among the biological agents used to mitigate the effects of mycotoxins, several studies have demonstrated their efficacy. Notably, PPAR-γ agonists [11,12], quercetin [13], resveratrol [14], sodium butyrate [15], and EPA and DHA [16] have shown novel applications in counteracting the harmful effects induced by mycotoxins. As widely described, peroxisome proliferator-activated receptor gamma (PPAR-γ) is a member of the PPAR subfamily [17], and it has been shown to regulate liver alterations (e.g., cardiac fibrosis [17]). Recently, some authors have shown how PPAR-γ can exert a protective effect against mycotoxins by expression of the transforming growth factor beta 1 (TGF-β1), thus alleviating the progression of cardiac fibrosis induced by these toxic agents [12].

Regarding quercetin, its potential to counteract mycotoxins effects relies on its antioxidant and free radical scavenging properties against DON, which have been recently demonstrated [13,18]. On the other hand, sodium butyrate, a histone deacetylase (HDAC) modulator, has been demonstrated to alleviate mycotoxin-induced liver injury via epigenetic modification [15].

As outlined by Seethaler et. al, long-chain n-3 polyunsaturated fatty acids (PUFAs), including EPA and DHA, have been shown to improve intestinal integrity in both animal studies and clinical trials [19]. An in vitro study conducted by Xiao et al. [16] showed that these fatty acids can promote cell growth, thus improving intestinal barrier function and, at the same time, downregulate the protein expression of necroptosis factors (such as tumor necrosis factor receptor TNFR1).

However, phytogenics—plant-derived products such as essential oils, herbs, and spices—are the most included substances in animal diets due to their potential to mitigate mycotoxin contamination. These phytogenic compounds exhibit antifungal, antioxidant, and immune-modulating properties that can reduce the incidence of mycotoxins and alleviate their negative effects on animal health and productivity [20,21]. Incorporating phytogenics into animal feed not only enhances feed quality and safety but also promotes sustainable, natural solutions to combat mycotoxin contamination.

### 1.1. Mycotoxins in Animal Feed: Effects on Animal Health and Production

Mycotoxicosis, resulting from mycotoxin exposure through inhalation, ingestion, or direct contact, can severely impact animal health. While it is important to control acute toxicity, it is equally crucial to monitor low-level chronic toxicity, as this can eventually lead to significant disruptions in animal health and production [8]. The co-occurrence of multiple mycotoxins in feed can exacerbate their harmful effects due to synergistic interactions, causing more severe damage than a single mycotoxin alone [8]. However, further research is needed to fully understand the impact of co-contamination and co-exposure to multiple mycotoxins in livestock feed. Currently, most studies focus on the effects of individual mycotoxins, leaving gaps in our knowledge regarding their combined impact on animal health.

Firstly, AFs, produced by the genera *Aspergillus*, are mycotoxins characterized by rapid absorption and metabolization into their active or detoxified metabolites through the liver [3]. These mycotoxins are known for their genotoxic, hepatotoxic, mutagenic, and carcinogenic effects, impacting reproduction and immune systems in both humans and animals, including rodents, poultry, swine, and ruminants. This results in liver damage, decreased productivity, poor eggshell and carcass quality, and increased susceptibility to various diseases [3,22,23,24]. Notably, aflatoxin B1 (AFB1) is widely known as the most toxic AF, linked to liver damage and diseases in both animals (rodents, livestock, and aquaculture species) and humans, including hepatocellular carcinoma [25,26]. Some species are described as the most susceptible to AFB1, such as turkeys, rats, pigs, sheep, and dogs, whereas others such as monkeys, mice, and chickens are considered resistant [27].

OTs, also produced by the *Aspergillus* and *Penicillium* species, primarily affect the kidneys, causing nephrotoxicity, kidney enlargement, and tumors in the urinary tract in animals (pigs, poultry, and rodents) and humans [28,29]. They can also disrupt the immune system and fetal development [22]. Ochratoxin A (OTA) is described as dangerous due to the carry-over of this mycotoxin into animal products from swine, poultry, and even ruminants [30,31]. The presence of OTA in milk from ruminant species represents a concern for human health [28].

DON, a trichothecene mycotoxin produced by the *Fusarium* species, can reduce feed consumption and weight gain, due to symptoms like vomiting, diarrhea, and nausea. DON also causes ribotoxic stress and immune suppression. Among all livestock species, swine are the most susceptible species to the toxicity of DON [22]. On the other hand, contamination by T-2 and HT-2 toxins, which are also trichothecene mycotoxins, can lead to oxidative stress through increased lipid peroxidation, and can mainly trigger carcinogenesis, immune depression, neurotransmitter imbalances, hepatotoxicity, nephrotoxicity, weight loss, growth retardation, oral lesions, and reproductive disturbances in humans, poultry, and rodents [32,33]. As observed with DON, swine are among the most susceptible animals regarding the effects of the T-2 toxin [34].

In addition, FBs, also from the *Fusarium* species, are commonly described as sphingolipid metabolism disruptors as they inhibit ceramide synthase and the acylation of sphinganine and sphingosine, a biomarker positively correlated to hepatotoxicity, nephrotoxicity, and carcinogenicity [35]. Additionally, FBs can restrict the mitochondrial electron transport chain complex I and promote the formation of reactive oxygen species (ROS), lipid peroxidation, and oxidative stress [36]. FBs affect numerous species, such as horses, pigs, sheep, cattle, fish, poultry, nonhuman primates, mink, rabbits, and rodents. However, natural disease outbreaks primarily occur in horses and pigs, with horses showing greater susceptibility [37].

ZEN, which is also produced by the *Fusarium* species, is characterized as an estrogen-receptor agonist that can trigger negative consequences, such as alterations in the reproductive tract, fertility impairment, hypoestrogenic syndromes, reduced testosterone and spermatogenesis, and finally cancer in reproductive organs. Exposure to ZEN leads to signs of hyperestrogenism, most notably in pigs, though it can also occur in other animal species to a lesser degree [38,39].

Finally, the emerging and concerning *Alternaria* mycotoxins, of which the most studied are TeA, AME, and AOH, are described as genotoxic, cytotoxic, carcinogenic, and mutagenic toxins correlated to carcinoma colon cells in humans and esophageal cancer in mammalian cells. However, further research is necessary to truly understand the toxic potential and effects that Alternaria toxins can cause to animal and human health, especially given the advice of the European Food Safety Authority (EFSA) in 2012, concerning the high levels of these toxins found in many food products in Europe [40]. Overall, these mycotoxins significantly impact animal health and production, highlighting the need for attention in the livestock industry.

Food and agricultural organizations estimate that 60–80% of feed crops are infected by mycotoxins, with 20% of the crops exceeding the European Union’s (EU) legal food safety limits [41]. Thus, chronic mycotoxicosis in feed crops entails additional costs, such as increased costs for health and veterinary care, regulatory fees, and investment in research costs focusing on relieving the impact and severity of the mycotoxin problem. This has a multi-million economic impact on the livestock industry [42]. Therefore, developing and implementing solutions to control and monitor factors implicated in mycotoxicosis is highly important to alleviate the negative effects in animal production caused by mycotoxicosis. Factors influencing the production and contamination of livestock feeds and foods by mycotoxins are divided into physical factors (environmental conditions), chemical factors (fungicides and fertilizers), biological factors (fungal species, strain specificity, strain variation, and instability of toxigenic properties), and agricultural practices (harvesting, storage, transport, treatments, and sanitation, among others) [42].

### 1.2. Natural Anti-Mycotoxin Solutions

Currently, different strategies are implemented in farms and agriculture to prevent or mitigate mycotoxicosis, addressing economic losses and factors implicated in reduced animal productivity. These strategies include good agricultural practices such as early harvesting, proper drying, physical treatments, sanitation, proper storage, and insect management. Additionally, chemical and biological controls, such as antifungal treatments and the use of atoxigenic strains, are implemented alongside breeding programs involving genetic bioengineering. Finally, decontaminating livestock feed through the use of feed additives is also a key strategy [42,43].

Feed additives can be categorized into different approaches: biological adsorbents and detoxication techniques, enterosorbents (adsorbent minerals), and anti-mycotoxin dietary compounds [42,43]. Biological adsorbents include bacteria (such as lactic acid bacteria), organic binders (bacteria and yeast cell wall and micronized fibers), and fungal conidia that reduce mycotoxin bioavailability [1]. Other biological methods involve bacteria, fungi strains, yeasts, or bioactive materials like enzymes or polypeptides to biodegrade mycotoxins and alleviate their toxic effects [1,44,45].

Enterosorbents, made from mycotoxin-binding minerals such as bentonites, sepiolites, zeolites, diatomites, aluminosilicates, hydrated sodium calcium aluminosilicate (HSCAS), and activated carbons, prevent the intestinal absorption of mycotoxins [46]. Their effectiveness depends on properties like charge distribution, polarity, pore size, and surface area [47]. Additionally, they improve the stability of food matrices by enhancing texture, consistency, and moisture control, contributing to better overall palatability and food quality [47]. However, some enterosorbents, like clays, have a non-selective absorption mechanism that can reduce the bioavailability of essential dietary nutrients such as vitamins, amino acids, and minerals [48,49]. The effectiveness of these enterosorbents is also influenced by dosage, incubation medium, and pH conditions [49].

Natural anti-mycotoxin dietary interventions are under development to explore innovative solutions based on nutraceutical, bioactive, and phytogenic compounds with antifungal and beneficial effects against the effects of mycotoxins on animal and human health. In particular, phytogenic feed additives derived from plant materials such as flowers, seeds, leaves, bark, and roots, possess nutraceutical qualities like antimicrobial, antifungal, antiparasitic, and antiviral properties, as well as mycotoxin-binding capabilities [50,51]. Compared to synthetic or chemical solutions and clays, natural feed additives are less toxic, leave no residue, and are nutritionally beneficial for animals while preserving the bioavailability of nutrients [52].

Dietary interventions also include amino acids, minerals, hormones, vitamins, and a wide range of phytogenic compounds. Amino acids play an important role in regulating metabolic pathways and cellular functions. Their antioxidant capacity makes them suitable candidates to protect against mycotoxin-induced oxidative stress [53,54]. Moreover, the immune and physiological functions of amino acids offer promising approaches for animal protection. The most studied amino acids with anti-mycotoxin properties include L-arginine, L-glutamine, and L-carnitine [53,54,55]. On the other hand, minerals like zinc, manganese, and selenium, which possess antioxidant, anti-inflammatory, antibacterial, and antiviral properties, are crucial for maintaining the body’s homeostasis and regulating immune functions. Therefore, the use of dietary minerals is a promising strategy against mycotoxins’ negative effects on body homeostasis [56,57,58]. Hormones like melatonin offer protection against mycotoxicosis through antioxidant, anti-inflammatory, and anti-apoptotic effects [59,60]. Vitamins, especially C and E, have been studied for their ability to protect vital organs such as the liver and kidneys from mycotoxin-induced damage, including oxidative stress and DNA damage [61,62,63].

Moreover, phytogenic bioactive compounds are emerging as promising solutions for controlling the toxic effects of mycotoxins in food and feed. These compounds support essential physiological pathways and functions while offering protective properties. Phytogenics derived from plant products like essential oils, herbs, and spices, are increasingly recognized for their antifungal, anti-inflammatory, and antioxidant effects, which can inhibit the growth of mycotoxin-producing fungi and reduce contamination in feed [64,65]. They disrupt fungal cell walls and membranes, inducing autophagy and activating the antioxidant system, as well as activating hormone-dependent signaling pathways to prevent fungal spoilage [66]. Phytogenics also interfere with the biosynthesis of mycotoxins or degrade their structures [66]. Moreover, phytogenics exhibit strong anti-inflammatory qualities, which help mitigate the inflammatory responses in organs such as the liver, kidneys, and gastrointestinal tract, which are commonly affected by mycotoxins. This reduces tissue damage and improves overall organ function [65,67]. Additionally, their potent antioxidant activity neutralizes free radicals generated by mycotoxins, protecting cells from oxidative stress [65,68]. Therefore, by combining these actions, phytogenics enhance the resilience of animals to mycotoxin-induced toxicity, supporting better health and productivity.

The most well-known phytogenic compounds with anti-mycotoxin effects include phenolic compounds (polyphenols), terpenes, and nitrogen-containing compounds. Phenolic compounds are powerful antioxidants that modulate cellular redox status by eliminating free radicals, chelating metal ions, and promoting the activity of antioxidative enzymes [69]. However, phenol antifungal activity depends on the chemical structure of the polyphenol, aldehydes, acid groups, conjugated double bonds, and the length of CH [70].

On the other hand, terpenes, volatile compounds formed by isoprene molecules, act as radical scavengers from environmental stresses by absorbing free radicals or radiation through double bonds. They also exhibit antifungal properties due to their lipophilic nature, which allows them to enter cells and modify membranes that affect cell permeability and electrochemical potential, preparing them to combat fungi and mycotoxins. Terpenes are mainly found in plant extracts and essential oils [71,72].

Nitrogen-containing compounds, such as glucosinolates, alkaloids, and cyanogenic glucosides, have defensive roles in cells, with some, like alkaloids and glucosinolates, possessing antifungal properties that target enzymes crucial to fungal processes [66,73]. Their main target is the inactivation of critical enzyme reductases, acetate kinases, and oxidases implicated in fungal processes [74].

In addition to their antifungal and antioxidant properties, some phytogenic compounds are emerging as effective biosorbents for binding mycotoxins [75,76]. These biosorbents, made from by-products and waste materials in the food industry, are eco-friendly and economical, containing biomolecules like polysaccharides and proteins with functional groups that bind to mycotoxins and even heavy metals [77]. While research on new biosorbents for mycotoxin removal is increasing, further studies are needed to determine their effectiveness and characteristics. Therefore, given the growing demand for natural alternatives to antibiotics in animal production, phytogenic compounds are gaining widespread acceptance as promising solutions. The following sections will highlight the major biosorbents currently used in animal nutrition, their mechanisms in the adsorption process of mycotoxins, and their implications for animal health and productivity.

## 2. Phytogenic Biosorbents Used in Animal Nutrition

The contamination of animal feed with mycotoxins poses a significant threat to livestock health and productivity, resulting in significant economic losses in the agriculture sector. Traditional methods for mycotoxin control, such as chemical binders and decontamination processes, have shown limited effectiveness and can cause side effects. In recent years, anti-mycotoxin phytogenic solutions have gained attention as a promising alternative. These natural compounds not only neutralize mycotoxins but also offer additional health benefits. This section focuses on well-established phytogenics already incorporated into animal and human diets, sharing common signaling pathways like antioxidant and anti-inflammatory mechanisms that boost their antioxidant, anti-inflammatory, and antifungal activities. Additionally, this section explores these phytogenic anti-mycotoxin solutions and their potential impact on animal health and productivity, offering a comprehensive overview of their role in modern animal husbandry.

### 2.1. Curcumin Extract

#### 2.1.1. Composition and Main Characteristics

Curcumin ((E, E)-1,7-bis (4-hydroxy-3-methoxyphenyl)-1,6-heptadiene-3,5-dione) is the main active component of turmeric (*Curcuma longa L*), among other curcuminoids, and is classified as a polyphenolic pigment [78]. Traditionally used as a natural homemade remedy in India, curcumin is known for its anti-inflammatory, antitumor, antioxidant, free radical scavenging, and immune-enhancing properties [78]. The synthesis of curcumin has been thoroughly investigated and optimized, as explained in the study by Pabon [79]. The steps to synthesize curcumin are summarized in Figure 1. Its structure is based on two aromatic rings containing o-methoxy phenolic groups, connected by a seven-carbon linker with an α, β-unsaturated β-diketone moiety (Figure 1).

According to regulation 2021/551 of the European Commission [80], the maximum recommended curcumin extract content is limited to 15 mg/kg of feed, with Curcuma essential oils limited to 20 mg/kg for most animals, except calves, where it is limited to 80 mg/kg. Curcumin is a non-toxic dietary compound rapidly and efficiently metabolized and eliminated by mammals, without accumulating in the body [81]. Upon ingestion, curcumin undergoes various biological reactions, including oxidation, nucleophilic addition, hydrolysis, degradation, and enzymatic reactions [81]. However, as a limiting factor, curcumin has low water solubility and is chemically unstable, leading to poor absorption and rapid metabolism [82]. To improve its solubility, stability, and bioavailability, innovative approaches have been developed, such as using olive oil as a solvent [83] or encapsulating curcumin in nanoparticles or nanospheres [84]. Marchiori et al. demonstrated that 10 mg/kg of nano-encapsulated curcumin had similar effects to 30 mg/kg of free dietary curcumin, highlighting the potential of these technologies to improve curcumin’s pharmacokinetics and pharmacodynamics [85]. Therefore, it is important to consider how curcumin is administered and what form it takes, in order to predict its bioavailability and bioactivity.

**Figure 1 toxins-16-00434-f001:**
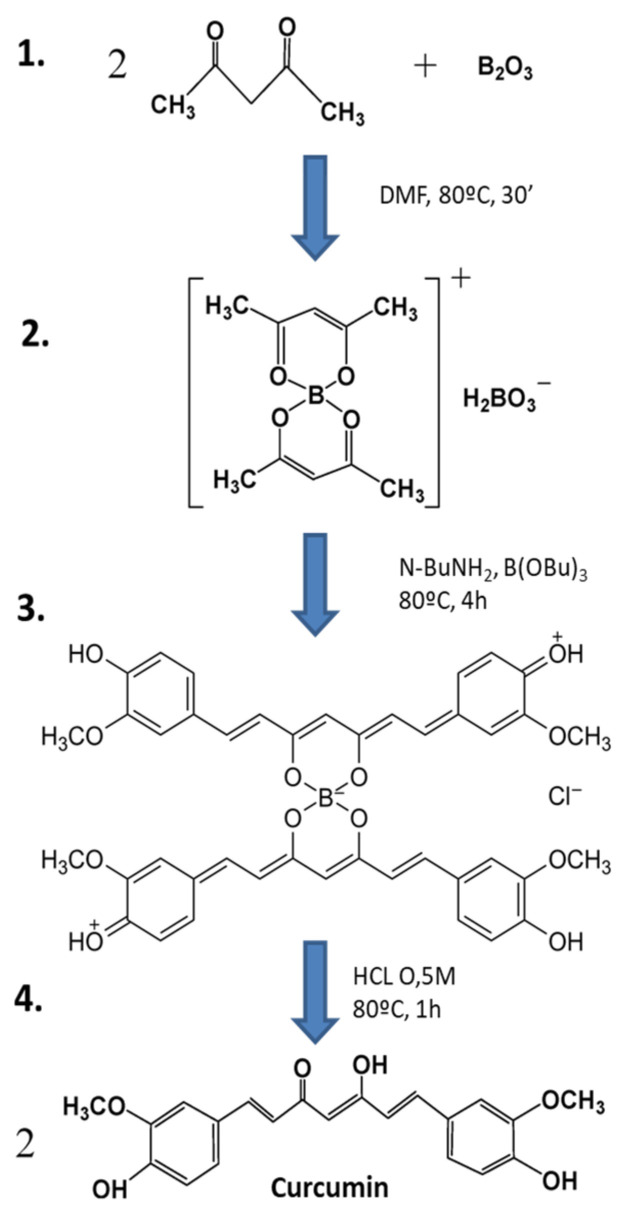
Synthesis of curcumin by the general method proposed by Pabon [79] (image adapted from Zerazion et al., 2016) [86]. **1.** A suspension of B_2_O_3_ (3 mmol) and acetylacetone (2 mmol) in 1.5 mL of DMF was stirred for 30 min at 80 °C. **2.** To this was added 4-hydroxy-3-methoxybenzaldehyde (5.4 mmol), followed by the slow addition of n-butylamine solution, and this was stirred at 80 °C for 4 h. **3.** The solution was acidified with 0.5 M HCl at 80 °C for 1 h. **4.** Curcumin was dried and recrystallized.

#### 2.1.2. Mode of Action and Implication on Animal Health and Production

Curcumin has three active sites that undergo oxidation, electron transfer, and hydrogen abstraction [78]. It acts as a reactive oxygen species (ROS) scavenger by transforming ROS into more stable phenoxyl radicals through hydrogen oxidation at the phenol-OH group of its three active sites [78,86]. As a result, curcumin protects cells from oxidative damage caused by ROS. Additionally, curcumin inhibits necroptosis and inflammation by blocking aflatoxin-activated RIP/MLKL and TLR4/NFκB signaling pathways [87]. The TLR4/NFκB signaling pathway inhibition also alleviates intestinal inflammation by lipopolysaccharide (LPS) [87]. Another benefit of curcumin is its ability to photosensitize and inactivate certain fungi and bacteria, reducing their levels [88]. Curcumin also acts as a liver protector and detoxifier by upregulating hepatic genes related to energy production, fatty acid metabolism, detoxification, coagulation, and immunological regulation [89].

Curcumin’s numerous biological effects in animals include anti-pulmonary fibrosis, anti-inflammation, anti-aging, antiviral, antibacterial, anticancer, immunomodulation, cardiovascular protection, and neuroprotection [90,91,92,93,94,95,96,97,98,99]. It affects key signaling pathways, such as AMPK, p53, p21, AKT, mTOR, Nrf2/ARE, NF-κβ, NLRP3, MAPKs, c-JUN, PPAR, and Wnt/β-catenin [100].

Studies have shown curcumin’s potential to inhibit the CYP450 enzyme, a key liver enzyme involved in activating aflatoxin B1 (AFB1), thereby protecting the liver and gastrointestinal tract [101,102]. Co-treatment with curcumin counteracts the harmful effects of ochratoxin A (OTA) by mitigating inflammation, reducing nitric oxide levels, and decreasing oxidative stress in the kidneys and liver in rats [103]. In addition, curcumin nanoparticles also help alleviate ZEN-induced hepatotoxicity in rats by ameliorating biochemical and cytogenetic parameters and the histopathological state [104].

In livestock, curcumin has shown significant protective effects against mycotoxicosis. For example, in male broiler chickens, curcumin [103] supplementation (400 mg/kg) reduced oxidative stress and damage in the liver and kidneys caused by 0.02 mg/kg of AFB1 [105,106]. Another study used 444 mg/kg of curcuminoids from turmeric or 300 mg/kg of curcumin with male 180-day-old broilers, demonstrating liver-protective effects and the ability to alleviate immune organ damage caused by the exposure to 1 mg/kg of AFB1 [87,107]. Moreover, the supplementation of 0.2% of curcumin acted as an immunomodulatory and protective agent against the damaging effects of AFB1 (2 ppm) on immune organs in male 151-day-old broiler chicks, such as the thymus, spleen, and bursa of Fabricius, and improved performance indicators like body weight, feed intake, and the feed conversion ratio [108]. Furthermore, 500 mg/kg of curcumin has been shown to protect the intestines of one-day-old broilers from 1 mg/kg of AFB1-induced damage by downregulating the Nrf2 signaling pathway [109]. Curcumin at doses of 150 mg/kg and 450 mg/kg alleviated liver and intestinal damage caused by 100 µg/kg of AFB1 in one-day-old broilers and 5 mg/kg of AFB1 in 120-day-old broilers, respectively, by downregulating CYP450 [101,110]. Additionally, 0.05% of turmeric extract can protect against the toxic effects of 3 ppm of AFB1 in the liver and kidneys of one-day-old broilers [111]. In several studies in one-day-old ducks, 400 and 500 mg/kg of curcumin demonstrated protective effects on the liver and intestine against acute damage induced by 0.06 and 0.1 mg/kg of AFB1. Curcumin suppressed the generation of CYP450 and AFB1-DNA adducts in the liver, and modulated gut microbiota, intestinal NLRP3 inflammasome, and the TLR4/NF-κβ signaling pathways in the intestines of the ducks [112,113]. The beneficial effects of curcumin extend to 28-week-old egg-laying hens and 45-week-old layer Japanese quails, where it has been shown to restore egg production and egg weight following AFB1 exposure (0.5 and 1.5 mg/kg, respectively) [114,115]. Additionally, 50 mg/kg of curcumin and 10 mg/kg of nano-curcumin demonstrated hepatoprotective and antioxidant properties in one-day-old male broilers intoxicated by FBs (600 mg/kg), minimizing the mycotoxicosis effect and increasing BW and ADG [116]. In another study, 400 mg/kg feed curcumin also provided partial protection against 1 mg/kg feed OTA in one-day-old broilers [117]. In addition, in broilers, the toxic effects of 0.5 ppm of OTA on kidneys were alleviated by 2 g/kg of feed curcumin [118]. In one-day-old ducks, OTA caused increased expression of apoptosis-related genes and the TLR4/ NF-κβ signaling pathway, and downregulated mitochondrial transcription factors A, B1, and B in the intestine. However, harmful effects induced by 2 mg/kg of OTA were diminished by 400 mg/kg of curcumin, protecting intestinal integrity and function, regulating gut microbiota, and improving performance parameters [90,119].

Curcumin has shown similar protective effects in swine, mitigating apoptosis and ROS in kidney cells contaminated with FB1, OTA, and DON [120]. A combination of curcumin and other phytogenics with adsorbent additives enhanced immune responses in pigs exposed to mycotoxins [121]. Furthermore, curcumin combined with silymarin and clays has improved reproductive and performance parameters in sows and piglets exposed to multiple mycotoxins. In 366-day-old sows, reproductive parameters and litter characteristics were ameliorated thanks to the supplementation of the multi-component mycotoxin-detoxifying agent and, in 28-day-old weaned piglets, BW and ADG were improved, and mortality and liver and intestinal damage were reduced significantly [122,123].

In an in vitro study with a bovine fetal hepatocyte-derived cell line (BFH12), curcumin mitigated the hepatotoxic effect of AFB1 by increasing antioxidant activity and the anti-inflammatory response. Antioxidant players, MDA concentration, and NQ01 enzyme activity were increased, and CYP3A activity, which plays a role in the bioactivation of AFB1, was reduced by curcumin administration [124]. In an in vivo study with dairy cows in their late lactation period, exposed to 5 µg/kg of AFB1, curcumin did not impact milk yield, milk composition, or somatic cell count. However, the low availability of curcumin in this formulation can be an issue in finding significant effects against mycotoxicosis, and better formulations are needed to maximize its influence on milk production [100]. Takiya et al. used a feeding phytogenic combination, in which curcumin and grape seed were used among other ingredients, in an in vitro study of dairy cows, and obtained benefits in performance parameters and nutrient utilization. Milk yield and fat were greater, and dry matter digestibility also increased [125].

In aquaculture, 400 mg/kg of curcumin alleviated OTA-inhibited myoblast differentiation and myofibrillar development-related proteins in a study with juvenile grass carp contaminated with OTA (1.2 mg/kg), ameliorating muscle toxicity and development, and enhancing growth performance mainly by increasing the weight gain percentage and final body weight. Additionally, Curcuma supplementation upregulated S6K1 and TOR, which are associated with the AKT/TOR signaling pathway, resulting in myoblast proliferation and fusion [126]. Also, 400 mg/kg of curcumin could alleviate 1.2 mg/kg OTA-induced gill and intestinal injuries in juvenile grass carp by reducing OTA residues in the gills and intestine, regulating TLR4 and NFκB signaling pathways, enhancing antioxidant defenses, decreasing oxidative stress, promoting a correct mitochondrial function, and decreasing necrosis and pro-inflammatory factors in the intestine [127,128]. In Red Tilapia, 50–60 mg/kg of nano-curcumin additive in feed increased disease resistance against *Aspergillus flavus* infection through the amelioration of hepatic damage and inflammation, intestinal dysfunction and morphology, splenic tissues and normal melano-splenic inflammatory macrophages, and improvement of the primary epithelium of the gill lamella and secondary filament in the gill tissue. This health amelioration in Red Tilapia improved growth performance parameters by increasing BW, ADG, FI, and fish survival, and decreasing FCR. Body composition was also ameliorated by nano-curcumin, reducing lipid content and increasing protein content [129]. El-Barbary demonstrated that the supplementation with curcumin in Nile Tilapia injected with AFB1 was effective in improving liver and kidney function and antioxidant activity by increasing the activity of detoxified enzymes and enhancing the detoxification of AFB1 through the suppression of CYP1A in the liver [130,131]. In addition, a diet of 5 g/kg of curcumin additive in feed for Nile Tilapia was effective against 2 ppm AFB1’s effects on growth performance, increasing BW, FI, ADG, and mortality rate, and reducing FCR. The residual AFB1 concentration was also reduced in Nile Tilapia by curcumin supplementation, which means curcumin can be a potential anti-AFB1 additive and protector against AF toxicity in this species [132]. In addition, the promising formulation of dietary Zn(II)-curcumin in shrimps exposed to AFB1 ameliorated hepatopancreas damage by conferring antioxidant and immunological protection [133].

In pets, 6–8 week-old rabbits aflatoxicated with 50 µg of oral AFB1 dissolved in 0.5 mL of olive oil were treated with 25 and 50 µg/kg of BW zinc oxide nanoparticles with curcumin, which conferred hepatoprotective effects by enhancing antioxidant activity, scavenging ROS, and improving hepatic condition [134].

Therefore, curcumin is a highly bioactive compound with significant potential for mitigating mycotoxicosis in livestock and pets. With effective doses ranging between 400 and 500 mg/kg in various studies, curcumin could play a crucial role in new strategies to combat the effects of mycotoxins in animal diets [101,119,135].

### 2.2. Silymarin Extract

#### 2.2.1. Composition and Main Characteristics

Silymarin (*Silybum marianum*) is the primary active compound extracted mainly from the seeds of the milk thistle, a thorny herb from the Asteraceae family. It has been widely studied for its potential in treating liver diseases [136,137]. The health benefits of milk thistle are largely attributed to silymarin, a complex composed of several flavonolignans such as silybin A, silybin B, isosilybin A, isosilybin B, silydianin, and silychristin (Figure 2). Additionally, silymarin contains flavonoids like quercetin and polyphenols. Among these, silybins are the most bioactive compounds, comprising 50–60% of silymarin, with the highest bioavailability. Despite its bioactive potential, silymarin is a highly hydrophobic molecule with low bioavailability, largely due to its rapid liver biotransformation and short half-life of six hours [138]. Beyond its hepatoprotective properties, guarding against liver diseases such as toxin-induced liver damage, viral hepatitis, cirrhosis, and hepatocellular carcinoma [139,140], silymarin has also demonstrated antioxidant, antifibrotic, anti-inflammatory, choleretic, immune-stimulating, regenerative, cytoprotective, cardioprotective, neuroprotective, and anti-carcinogenic effects [141].

Once ingested, both free and conjugated forms of silybin are rapidly metabolized by the body, with a half-life of six hours. Silybin is recognized as a foreign substance and, consequently, metabolized by phase II enzymes [142]. As with curcumin, silymarin is also a bioactive compound with low bioavailability and fast metabolism, and its concentration in plasma is also low, as it is mainly found in the conjugated form. However, upon consumption, silymarin induces reactions such as increased glutathione S-transferase (GST) and quinone reductase (QR) activity in certain organs, thanks to its antioxidant properties [142].

#### 2.2.2. Mode of Action and Implication on Animal Health and Production

Silymarin acts not only as a free radical scavenger but also inhibits enzymes involved in free radical production and maintains mitochondrial electron-transport chain integrity during oxidative stress [142]. It protects mitochondria from reactive oxygen species (ROS)-induced protein modifications, lipid peroxidation, and mitochondrial DNA damage by reducing electron leakage, ROS formation, and inactivating ROS-producing enzymes within the mitochondria [142].

Liver damage induced by various factors can be mitigated through the use of silymarin, a compound that reduces hepatic enzyme levels (ALT, AST, and ALP) by activating ribosomal RNA synthesis, promoting liver regeneration, and preventing fibrosis [143]. Notably, silymarin is capable of deactivating the CYP450 enzyme, which plays a crucial role in detoxifying mycotoxins, thereby reducing glutathione (GSH) consumption in the liver [143].

Given these antioxidant properties, silymarin has been tested for its ability to mitigate the effects of mycotoxins. It inhibits lipid peroxidation and enhances enzymes related to oxidative stress, such as catalase (CAT), superoxide dismutase (SOD), GSH peroxidase, GST, and GSH reductase. Additionally, silymarin protects organs from damage by activating the PI3K-Akt cell survival pathway, preventing apoptosis [144]. For instance, in a preclinical study with mice, silymarin alleviated liver damage caused by FB1 toxicity through its antioxidant effects, reducing the expression of vascular endothelial growth factor (VEGF) and fibroblast growth factor-2 (FGF-2), as well as the apoptotic rate, thereby conferring a hepatoprotective effect [145].

Silymarin also alleviated ZEN-induced hepatotoxicity and reproductive toxicity by reducing histopathological lesions in the liver, ovaries, and uterus. This effect was attributed to the mTOR signaling pathway regulation and the reduction of oxidative stress and the ZEN-related protein metabolism [146]. In another study, nano-encapsulated silymarin, designed to improve solubility and bioavailability, mitigated DON toxicity by counteracting elevated serum enzyme activity, cholesterol, triglycerides, LDL, cytokines, lipid peroxidation, and nitric oxide (NO), while enhancing antioxidant enzymes such as GPx, SOD, and CAT [147]. Additionally, silymarin protected against FB toxicity by preventing cellular damage, reducing lactate dehydrogenase production, and lowering Tnf-α expression [148].

In various livestock species, silymarin supplementation has demonstrated multiple health benefits. For example, in 14-day-old broilers, silymarin extract countered the adverse effects of 0.8 mg/kg of AFB1 on liver health, improving performance and meat quality [149]. In one-day-old broilers exposed to 250–500 ppb of AFB1, diets containing 0.5–1% of *Silybum marianum* seeds improved liver status and acted as a hepatoprotective agent [150]. Similarly, seven-day-old broilers affected by 2 ppm of AFB1 and supplemented with 1000 ppm of silymarin showed enhanced villi height and VH:CD ratio, improved absorptive surface area, reduced pathogenic bacteria growth, and ameliorated FI, ADG, and FCR [151]. Armanini et al. observed that 100 mg/kg of silymarin improved productivity, meat quality, oxidative stress, and polyunsaturated fatty acids (PUFAs) in broilers exposed to 0.05 ppm of AFB1 and 20 ppm of FBs [152]. Likewise, in a study with one-day-old broilers contaminated with 8.4 ppb of AFB1 and 24.3 ppb of T-2 toxin, 0.5% silymarin improved performance parameters such as BW, BWG, FI, and FCR [153]. In ducks, a 0.5% milk thistle seed supplementation alleviated liver damage from 4.9 mg/kg of DON and 0.66 mg/kg of ZEN by activating the antioxidant system [154]. Additionally, in one-day-old ducks fed with diets naturally contaminated with ZEN (0.46–0.50 mg/kg of feed) and DON (3.43–3.72 mg/kg of feed), 0.5% milk thistle seed, 0.5% milk thistle seed cake, or 0.1% milk thistle seed oil prevented histological alterations in the liver, spleen, and bursa of Fabricius [155]. Similar protective effects were observed in laying hens, where 1% silymarin supplementation reduced the impact of 1 mg/kg of OTA on performance, increased egg production and quality, and protected the liver and kidneys from oxidative stress [156]. In a study with one-day-old chicks intoxicated with 3 mg/kg of OTA, 1000 mg/kg of silymarin protected the liver and kidneys against oxidative stress induced by mycotoxins [157]. In quails fed with a diet contaminated with 2.2 mg/kg of AFB1, 2000 mg/kg of silymarin alleviated the adverse effects of AFs on performance parameters such as FCR and ADG, and on hepatic function serum biomarkers [158].

In nursery pigs contaminated with 500 ppb of AFB1, a blend of silymarin and other phytogenics protected the liver and intestines from mycotoxin-induced damage, improving liver biomarkers and gut morphology [159]. Additionally, in 25-day-old post-weaned pigs exposed to 0.35 mg/kg of ZEN and 0.5 mg/kg of T-2 toxin, silymarin supplementation reduced mycotoxin residues, enhanced biochemical parameters, and improved growth performance [160]. Similar results were found in piglets exposed to 0.992 mg/kg of ZEN and 0.531 mg/kg of OTA and supplemented with a binding agent combined with silymarin in-feed, improving FCR [161]. Another mycotoxin deactivator blend with silymarin caused an amelioration in the performance parameters by increasing BW, ADG, and FI, and reduced oxidative stress in 24-day-old weaned pigs contaminated with 4.5 mg/kg of DON, 0.5 mg/kg of ZEN, and 18 mg/kg of FBs [162]. As aforementioned, the combination of curcumin extract and silymarin extract with a mix of selected clays to test a novel mycotoxin detoxifier formulation showed protective and ameliorative effects on weaned piglets and sows fed with a diet naturally contaminated with multiple mycotoxins [122,123].

Finally, 600 mg/kg of silymarin supplementation in dairy cows intoxicated with 1 mg/kg of AFB1 mitigated AFB1’s toxic effects, improving renal and liver function, and increased ADG and FI [163]. Additionally, 600 mg/kg BW of silymarin phytosome supplementation in dairy cows exposed to 0.8 mg/kg of AFB1 succeeded in reducing AFM1 carry-over in milk [164]. On the other hand, the supplementation of silymarin succeeded in reverting AFB1 toxic effects, ameliorating rabbit health [165].

Given these findings, silymarin is a promising ingredient for mitigating mycotoxicosis in livestock animals. Some studies suggest that combining silymarin with adsorbent clays may further enhance its detoxifying properties [122,123,166]. To note, the maximum effective dosage of silymarin across livestock studies typically ranges from 5–10 g/kg of diet.

### 2.3. Grape Pomace Extract

#### 2.3.1. Composition and Main Characteristics

Grapes (*Vitis vinifera* L.) are the most widely cultivated fruit crop globally, and they are used in the food industry to produce various products, such as wine [167]. During the grape processing for wine production, by-products are generated that possess significant nutritional value. Circular economy practices have opened new avenues for reusing these by-products in various applications, including the development of new food products or nutraceutical ingredients. These ingredients, with their added value, are beneficial for both human and animal consumption [168]. The by-products from wine production, collectively referred to as grape pomace, consist of grape seeds, grape skins, and grape stalks. However, using these by-products directly from waste can introduce toxins and other harmful substances along with health-beneficial compounds [108]. Therefore, it is essential to develop efficient extraction techniques to optimize the recovery of by-products with the best biological and physico-chemical characteristics [168,169].

Grape pomace contains a rich variety of beneficial compounds, including unsaturated lipids, sterols, vitamins, antioxidants, fiber, volatile organic compounds and, most notably, phenolic compounds. These phenolic compounds are of particular interest due to their beneficial properties, which include reducing oxidative stress and inflammation, and even possessing anticancer properties [170]. The most significant phenolic compounds found in grape pomace include simple phenols like hydroxycinnamic and hydroxybenzoic acids, as well as polyphenols such as tannins, stilbenes, and flavonoids [171]. Additionally, the phenolic profile can vary depending on the grape variety, which plays an important role in the selection process [171]. In grapes, phenolic compounds are divided into two main categories: non-flavonoids and flavonoids. Flavonoids, characterized by a C6–C3–C6 structure, are prevalent in grapes and are composed of two hydroxylated benzene rings (A and B) linked by a tricarbon chain that forms part of the heterocyclic ring C. These compounds are further categorized into different structural classes based on the oxidation state of ring C, including flavonols, flavan-3-ols (encompassing simple flavan-3-ols and their polymeric form, proanthocyanidins), and anthocyanins [172] (Figure 3). On the other hand, the most abundant non-flavonoids in grapes are hydroxycinnamic acids, such as caftaric acid and coutaric acid [172].

Despite the wide range of health benefits attributed to polyphenols, it remains unclear which specific compounds contribute the most to these effects. Polyphenols must be released from the food matrix and transformed into bioaccessible forms in the gastrointestinal tract to be absorbed into the bloodstream. Most dietary phenolics are poorly absorbed in the small intestine and undergo extensive metabolism by the gut microbiota in the large intestine, producing metabolites that differ structurally from the original compounds [173]. As a result, dietary phenolic compounds generally have low bioavailability, with phenolic acids being the most bioavailable and anthocyanins the least. Anthocyanins, which are primarily found in red grapes, act as natural colorants and are highly stable, making them difficult to digest and absorb [170,174]. In contrast, proanthocyanidins, which are the most abundant phenolic compounds in grape seeds, exhibit better bioavailability [175]. In conclusion, most phenolic compounds in grape pomace are not absorbed in the small intestine, with 90–95% remaining unabsorbed. While some phenolic compounds are directly available, the majority are partially broken down by intestinal enzymes and microorganisms. The resulting metabolites from polyphenol digestion are then absorbed and utilized by the body through passive diffusion or active transport [170]. Regarding the use of dry grape extract in animal nutrition, EFSA has conducted a study on the safety and efficacy of this feed additive. The study concluded that a maximum dosage of 100 mg/kg of feed for animal species poses no safety concerns and is not considered a risk to the environment [176].

Flavonoids are high-quality bioactive molecules with natural antioxidant properties, and hold great potential for use in the food industry. In addition to their antioxidant activity, flavonoids possess anti-inflammatory, estrogenic, antimicrobial, antidiabetic, and antitumor properties, making them valuable in the development of nutraceutical strategies for combating diseases and as functional food additives. Grape pomace, specifically, has been shown to inhibit the growth of certain harmful bacteria, further enhancing its value [156,177].

#### 2.3.2. Mode of Action and Implication on Animal Health and Production

The molecular mechanisms by which flavonoids and non-flavonoids exert their effects are diverse [178]. The antioxidant properties of polyphenols, particularly proanthocyanidins, are associated with the prevention of lipid oxidation. Studies have shown that proanthocyanidins reduce oxidative stress by lowering levels of lipid peroxidation and Nε-carboxymethyl lysine (products of reactive oxygen species activity), and decreasing the activity of antioxidant enzymes such as SOD and GSH in a diabetic preclinical model [170]. Anthocyanidins, flavonols, and other phenolic compounds present in red grape pomace have been found to decrease ROS levels, TBARS, and carbonyl levels, while increasing GSH levels by promoting the enzymatic activation of glutamyl cysteine synthetase (GCS) and glutathione S-transferase (GST) in muscle cells [179,180]. In cardiovascular diseases, grape pomace rich in proanthocyanidins, anthocyanins, and quercetin has been shown to restore the expression of genes such as eNOS, SOD2, and HO-1, which are involved in reducing hypertension and providing vascular protection [181]. In a preclinical study, grape pomace supplementation has also been found to increase the activity of antioxidant enzymes such as SOD, GPx, and glutathione reductase in models of colitis [182].

The multiple benefits of grape pomace have sparked interest in its use as a feed additive in animal nutrition. Grape pomace has demonstrated its ability to act as a biosorbent, capable of adsorbing mycotoxins by hydrophobic interactions such as AFB1 and ZEN, and forming polar non-covalent interactions with OTA and FB1 [183]. Additionally, grape pomace has been observed to exhibit biofungicidal properties against key mycotoxigenic species [184]. As a result, grape pomace presents a cost-effective solution for decontaminating mycotoxin-contaminated feed and offers antioxidant and antimicrobial benefits, making it a promising anti-mycotoxin product. For consideration, the maximum effective dosage of grape extract observed in different studies with livestock animals was around 80–100 g/kg of diet.

Studies on livestock animals have further demonstrated the effectiveness of grape seed extract in mitigating the harmful effects of mycotoxins. In one-day-old broiler chickens, 250 and 500 mg/kg of grape seed extract reduced the oxidative stress in the liver and plasmatic cytokine levels enhanced by 400 mg/kg of FB1. Grape seed extracts also ameliorated jejunal morphology and ileal microbiota homeostasis, and improved BW, FCR, and mortality rates, resulting in better growth performance [177]. Additionally, grape seed extract counteracted AFB1’s effects on the immune function, antioxidant capacity, and liver morphology of one-day-old broiler chickens by reducing AFB1 residues and damage in the liver, resulting in increased serum immunoglobulin levels and antioxidant parameters and improved growth performance, by improving the organ’s relative weight, ADG, and FCR [37,185,186]. Moreover, 200 mg/kg of grape seed extract can improve the immune function of one-day-old broiler chickens by counteracting the influence of 76 ppb of AFB1 in the growth of the thymus gland and the relative weight of the bursa of Fabricius [187]. In addition, 2 mL/L of gallic acid extracted from grape skin improved hematology, serum biochemistry, and histological morphology in the lungs, liver, and ileum in 100-day-old broilers contaminated with 2.7 × 10^6^ spores/mL of *Aspergillus flavus* [188]. Similar improvements were observed in quails when supplemented with 500 mg/kg of grape seed extract, including relief of the negative effects induced by 1 mg/kg of AFB1, such as oxidative stress, lipid peroxidation, liver fibrosis, and sinusoidal dilation. In addition, grape seed extract improved performance parameters in quails such as FCR, BW, and BWG [189].

Furthermore, the supplementation of grape seed extract in a study with piglets exposed to 479 ppb of OTA and 62 ppb of AFB1 reduced portal areas with mononuclear cellular infiltration and periportal fibrosis in the liver, reduced atrophy of the glomerular tufts, and altered the Bowman’s capsule. In this regard, CYP450 expression in the liver was downregulated by grape seed extract resulting in the bioinactivation of these mycotoxins [181]. In a study with piglets fed with a bolus contaminated with FB1, ZEN, OTA, DON, and AFB1, supplementation with grape pomace reduced the gastrointestinal absorption of these mycotoxins; therefore, the mycotoxicosis of these piglets was reduced [190,191]. Moreover, 8% grape seed extract reduced oxidative stress and inflammatory markers in the mesenteric lymph nodes, spleen, and liver of four-week-old weaned piglets exposed to 320 μg/kg of AFB1 [192,193]. Similarly, cross-bred TOPIGS-40, contaminated after weaning with 320 ppb of AFB1, were given supplements with 8% grape seed extract, enhancing antioxidant enzyme activity and reducing pro-inflammatory cytokines and TBARS levels [194]. In addition, 8% grape pomace feeding in cross-bred pigs (TOPIG-40) exposed to 320 g/kg of AFB1 positively improved BW and kidney status by boosting the tissue’s antioxidant status [195]. Regarding microbiota homeostasis, 8% dietary grape seed administration to four-week-old piglets contaminated with 320 g/kg feed AFB1 caused an increase in the concentration of beneficial microbiota in the large intestine, causing a reduction in diarrhea and an increase in BW [196].

Furthermore, the use of grape pomace in aquatic animals has also shown positive results. For example, the administration of 1 g/kg of tannins to juvenile Chinese sea bass improved intestinal integrity and bacterial homeostasis, leading to enhanced growth performance and a resistance to AFB1 toxicity [197]. Similarly, 0.2% quercetin, another polyphenol found in grape pomace, has been shown to protect against liver and muscle damage in tilapia exposed to 4.8 mg/kg feed T-2 toxin, improving survival and performance parameters [198]. Therefore, quercetin was effective in protecting the liver and the muscle from T-2 toxicity and in enhancing performance parameters.

### 2.4. Olive Pomace Extract

#### 2.4.1. Composition and Main Characteristics

During the olive oil extraction process, by-products rich in bioactive compounds are generated. Since olive oil production yields only 15–20% of the raw material, the remaining 80–85% consists of olive pomace [199]. Olive pomace is composed of sugars, fibers, polyalcohols, volatile fatty acids, pectins, fats, and various phenolic compounds like hydroxytyrosol, tyrosol, secoiridoid derivatives, phenolic acids, and flavonoids [200]. Olive pomace has well-known health benefits for both animals and humans. Similar to grape extracts, olive pomace is a by-product of olive oil processing, making it an eco-friendly practice within the circular economy, adding value to the olive oil industry [201]. Furthermore, the bioactive compounds in olive pomace have attracted significant interest in the pharmaceutical and functional food industries.

Olive pomace contains crushed olive stones, pulp, skins, and varying amounts of water, depending on the oil extraction method used. It is rich in phytochemicals, including vitamin E vitamers, phenolic compounds, peptides, quercetin, and fats (Figure 4) [202]. The most important vitamin E vitamer is α-tocopherol, the most abundant and biologically active form of vitamin E present in olive pomace. It plays a crucial antioxidant role by scavenging ROS and preventing lipid peroxidation. Apart from α-tocopherol, olive pomace contains seven other important vitamers: β, δ, and γ-tocopherols, and α, β, δ, and γ-tocotrienols [203].

In terms of fats, olive pomace is rich in monounsaturated fatty acids (MUFAs) and contains polyunsaturated fatty acids (PUFAs). The high MUFA content makes olive pomace less susceptible to oxidation. Oleic acid (C18:1n9cis), the most abundant MUFA in olive pomace, has numerous health benefits for both animals and humans. Additionally, olive pomace contains linoleic acid, which has antibacterial properties [203,204].

Phenolic compounds are another significant bioactive component of olive pomace, with hydroxytyrosol being the most abundant, followed by tyrosol and oleuropein [205]. Hydroxytyrosol, the main product of oleuropein hydrolysis, is one of the most potent natural antioxidants found in olive pomace. It exhibits antioxidant, antitumor, anti-inflammatory, and antimicrobial properties, and it helps regulate lipid profiles while protecting against genotoxicity, cytotoxicity, and proapoptotic effects [200]. Oleuropein, another important secoiridoid phenolic compound, provides antioxidant and anti-inflammatory properties due to the hydroxyl groups in its structure, which donate hydrogen to prevent oxidation. Moreover, oleuropein has been studied for its ability to modulate blood pressure, helping to prevent hypertension [206]. Additionally, oleuropein has demonstrated anti-cancer, cardioprotective, neuroprotective, and lipid-regulating effects in metabolic disorders [200,205]. On the other hand, tyrosol, a phenylethanoid compound, exhibits antimicrobial, anti-carcinogenic, anti-atherogenic, anti-inflammatory, and antioxidant properties [200,205].

Despite these favorable characteristics of phenolic compounds in olive pomace, their bioavailability and bioaccessibility are low due to their incomplete intestinal absorption and fast biotransformation during digestion, ending up in the excretion of polyphenols from olive pomace [207,208]. During gastrointestinal digestion, hydroxytyrosol and oleuropein concentrations remain stable whereas the bioaccessible tyrosol significantly increases compared to the native and undigested sample. However, the bioaccessibility of these polyphenols can be improved by innovative formulation approaches, such as the use of pharmaceutical excipients and advanced formulations for oral administration [208] or through the fermentation of olive pomace [209]. EFSA has found extensive scientific evidence supporting the health benefits of hydroxytyrosol and related polyphenols [210]. Furthermore, EFSA has raised no safety concerns regarding their use in novel food preparations at doses up to 215 mg/kg for humans, and hydroxytyrosol showed no adverse effects in rats at 2000 mg/kg of body weight in acute toxicity studies and 500 mg/kg of body weight in chronic repeated doses [40,210].

#### 2.4.2. Mode of Action and Implications for Animal Health and Production

Olive pomace, rich in valuable bioactive compounds, is increasingly used in developing functional food ingredients and nutraceuticals. Compounds like hydroxytyrosol, oleuropein, and quercetin exhibit anti-cancer activity by promoting apoptosis in cancer cells, thus reducing their viability [211,212,213]. The antioxidant properties of polyphenols in olive pomace stem from their hydroxyl groups, which neutralize reactive oxygen and nitrogen species, making them effective hydrogen donors. Additionally, they can chelate metal ions to reduce free radical production. These hydroxyl groups also disrupt bacterial cell membranes, contributing to antimicrobial activity [203]. Olive pomace has demonstrated high antioxidant capacity, effectively reducing ROS levels in cancer cells [213]. It increases glutathione (GSH) levels, protecting cells from radical-induced damage. In fact, hydroxytyrosol has shown the highest intracellular radical scavenging activity in pre-treated cells. Olive pomace also reduces glutathione peroxidase (GPx) activity, altering the cells’ redox state through direct radical scavenging [214]. In vascular diseases, olive pomace exhibits anti-atherogenic effects by decreasing pro-atherogenic factors (iNOS, COX-2) and pro-inflammatory mediators (TNF-α, MMP-2, MMP-9, and MT1-MMP), reducing nitric oxide (NO) production, and improving vascular health [206]. Hydroxytyrosol also downregulates the expression of the vascular cell adhesion molecule (VCAM-1) and the intracellular cell adhesion molecule (ICAM-1), both of which are involved in inflammation and are activated by NF-κβ [215]. Olive pomace further exerts anti-inflammatory effects by upregulating TIMP-1 and inhibiting the NF-κβ pathway, which modulates TNF-α and matrix metalloproteinases (MMPs). In a study involving one-day-old broilers, olive pomace reduced gut inflammation by downregulating IL-8, a pro-inflammatory cytokine, and upregulating the anti-inflammatory cytokine TGF-β4 [216]. Additionally, polyphenols in olive pomace were linked to inhibiting the activation of p38 and ERK1/2 signaling pathways, further contributing to its anti-inflammatory effects [217].

Regarding mycotoxin mitigation, studies have shown that olive pomace can inhibit the growth of toxigenic fungi and reduce levels of AFs [218]. Moreover, an in vitro study showed that olive pomace potentially adsorbs AFB1, ZEN, and OTA mycotoxins at 30 mg/mL at different pH levels [219]. Specifically, hydroxytyrosol has exhibited protective effects against oxidative stress induced by OTA in kidneys from rats, by limiting the release of MDA, lactate dehydrogenase (LDH), and ROS [220]. Olive leaves and branches have also been found to possess anti-mycotoxin biosorbent properties [221], and olive mill wastewater has shown an inhibitory effect on AFB1 production [222].

Quercetin found in olive pomace, as aforementioned in the grape pomace section, has shown protective effects against T-2 toxin-induced damage in tilapia by enhancing antioxidant capacity, protecting liver function, and improving survival rates, weight gain, and the hepatosomatic index [198].

As a low-cost biosorbent, olive pomace can be used to remove mycotoxins from contaminated animal feed. It also holds promise as a feed additive, offering health benefits that enhance the value of anti-mycotoxin products. However, further studies are needed to understand better the beneficial effects of olive pomace on animal health, especially concerning mycotoxicosis. Notably, the maximum effective dosage of olive pomace extract observed in various livestock studies is around 5 g/kg of diet.

### 2.5. Orange Peel Extract

#### 2.5.1. Composition and Main Characteristics

For a long time, oranges (*Citrus sinensis*, *Citrus × aurantium*, or other varieties) were primarily used in the food industry for human and animal consumption, as well as for juice production. This process generated significant amounts of waste, including orange seeds and peels, which typically went untreated or were discarded. However, the food and animal industries are now recognizing the beneficial health properties of bioactive compounds found in orange peel, which contains a phenolic concentration more than 15% higher than the pulp. These compounds can be utilized in new feed additives and nutraceutical formulas. As a result, orange peel has gained renewed value, aligning with circular economy practices by repurposing orange by-products, thanks to the promising health benefits its components offer [223,224].

Orange peel contains several phytochemicals of high interest due to their health-promoting and disease-preventing properties. These include vitamin C, folic acid, potassium, pectin, polyphenols, volatile compounds, minerals, dietary fibers, essential oils, and carotenoids. The outer layer, known as the flavedo, is rich in carotenoids and essential oils, while the inner layer, called the albedo, is rich in pectin and phenolic compounds. The phenolic compounds in orange peel include both non-flavonoids and flavonoids, such as polymethoxylated flavones (PMFs), C- or O-glycosylated flavones, O-glycosylated flavanones, and flavonols [87,225,226]. The most abundant citrus flavonoids in the orange peel are neohesperidin, hesperidin, naringin, nobiletin, narirutin, didymin, sinensetin, eriocitrin, 3′, 4′, 5, 5′, 6, 7-hexamethoxyflavone, diosmetin, and tangeretin [227].

The main health benefits attributed to the phenolic compounds in orange peel include antioxidant, anti-inflammatory, anticancer, anti-proliferative, and antiviral activities. They also contribute to improved capillary fragility and the prevention of platelet aggregation, helping to mitigate various chronic diseases [224]. Glycosylated flavanones, such as hesperidin, naringin, and neohesperidin, are the most abundant polyphenols in orange peel. These polyphenols have shown significant antioxidant and anti-cancer effects in multiple studies, as they stabilize free radicals by donating protons or electrons. However, their oral bioavailability is limited due to the conjugation of their hydroxyl groups and low water solubility, leaving many of the polyphenols unabsorbed and persisting in the colon [59,226]. Despite this low bioavailability, some strategies may enhance this characteristic through enzymatic modifications or the design of delivery systems [109].

In addition to polyphenols, orange peel is a rich source of carotenoids. They are lipophilic non-phenolic compounds that act as natural pigments and offer health benefits through antioxidant and pro-vitamin A activities [228]. Carotenoids are precursors to vitamin A (retinol) that play a critical role in metabolism and chronic disease prevention [229]. These compounds are classified into carotenes, which contain only carbon and hydrogen atoms, and xanthophylls, which contain oxygenated groups, such as hydroxyl, epoxy, and carboxyl groups [225]. The most abundant carotenoids in orange peel include β-carotene, lycopene, isolutein + zeaxanthin, violaxanthin, antheraxanthin, lutein, and β-cryptoxanthin, among others (Figure 5). Carotenoids have been associated with reducing the risk of cancer, osteoporosis, age-related muscular degeneration, cataracts, sunburn-induced skin damage, and cardiovascular diseases [230]. On the other hand, pectin, which is a non-phenolic soluble fiber, appears to ameliorate digestive complications, reduce cholesterol levels, and act as a protective agent against cancer and cardiovascular diseases [231]. Nevertheless, the presence of pectin and its lipophilic characteristics may interfere with carotenoid bioaccessibility. Adjustments to the molecular structure of carotenoids or encapsulation techniques may improve their bioaccessibility and bioavailability in the gastrointestinal tract [232].

#### 2.5.2. Mode of Action and Implication on Animal Health and Production

The health benefits of the bioactive compounds in orange peel depend on their molecular structures and mechanisms of action. For example, hesperidin and neohesperidin, key flavonoids in orange peel, exhibit radical-scavenging activity [231], suppress ROS and IL-5 via Nrf2 protein expression, activate the ERK/JNK and PI3K/Akt signaling pathways, and induce HO-1 protein expression by upregulating the transcription factor PPARγ [233]. Hesperidin is especially effective as a ROS scavenger and metal-chelating agent [234,235]. Additionally, hesperidin supports endothelial cell integrity, which is vital for proper vascular function [235]. Hesperidin also shows potential as a feed additive for poultry, as it reduces pro-inflammatory cytokine production and mitigates the effects of influenza A (H1N1) by suppressing MAPK signaling pathways and upregulating p38 and JNK expression, which enhances cell-autonomous immunity [229,230].

Naringin, another key flavonoid, reduces oxidative stress, lowers cholesterol levels, increases SOD activity, and promotes hepatic depuration of LDL cholesterol in rats. This results in improvements in metabolic disorders [236]. Naringin also regulates pro-inflammatory markers such as IL-6, IL-8, iNOS, Nrf2, COX-2, and TNF-α, acting as a potent anti-inflammatory agent in rats [237]. Importantly, naringin functions as an anticancer agent by inhibiting cell proliferation, triggering apoptosis, and diminishing the migration and invasion of specific tumor cells, thereby influencing cell motility [237]. According to several studies in human cells, naringin inhibits cell proliferation, triggers apoptosis, and decreases tumor cell migration and invasion. It achieves this by interrupting EGFR signaling and halting the cell cycle at the G0/G1 phases, as well as by inhibiting VEGF, FAK (PTK2), MMPs, and Zxb1, all of which play roles in tumor development [238].

One of the main carotenoids found in orange peel, β-cryptoxanthin, has been shown in mice to inhibit pro-inflammatory cytokines such as TNF-α, IL-1β, IL-6, and TGF-β1, as well as collagen type Iα1 and plasminogen activator inhibitor-1 [239]. Additionally, β-cryptoxanthin has protective effects against DNA damage and lipid peroxidation in humans [240]. It also enhances alkaline phosphatase activity and calcium content in the metaphyseal tissue and cortical bone of rats, thereby protecting against osteoporosis by inhibiting bone resorption [241].

Pectin, another important component of orange peel, has numerous health benefits for gut microbiota and gut health. It enhances beneficial bacterial populations, improves digestive and liver enzyme activities, and promotes nutrient absorption. Pectin also serves as an immunomodulator by stimulating the production of antibodies and macrophages [242], making pectin a promising feed additive for improving livestock productivity.

Furthermore, in combination with clays, orange peel has shown enhanced stability in the gastrointestinal tract, increasing the adsorption of mycotoxins such as AFB1, OTA, and FB1 in in vitro studies [243]. Orange peel has also demonstrated the ability to sequester AFB1, OTA, and ZEN [221]. Additionally, orange peel extract exhibited antifungal properties against *Aspergillus flavus*, a pathogenic fungus linked to various mycotoxicoses in the feed industry [244]. For instance, in an in vitro study, orange peel extract efficiently adsorbed 90% of the AFB1 under gastrointestinal tract conditions [245].

Dietary lycopene (200 and 400 mg/kg), a carotenoid found in orange peel extract, has been shown to alleviate 100 µg/kg of AFB1 toxicity in the gut and the liver of one-day-old broilers by enhancing antioxidant capacity and function and improving inflammatory status, leading to better intestinal health [246]. Lycopene also inhibited hepatic CYP450, which is responsible for liver damage [246,247]. Moreover, 200 mg/kg of dietary lycopene improved growth performance and meat quality in one-day-old broilers exposed to 100 µg/kg of AFB1 [248]. Additionally, dietary lycopene alleviated the toxic effects of DON and ZEN in swine by activating the OXPHOS signaling pathway and enhancing mitochondrial function in intestinal epithelial cells [83,249]. Lycopene also promoted Nrf2 pathway expression, reducing autophagy and apoptosis, thereby mitigating oxidative damage to Sertoli cells caused by ZEN [250].

Overall, polyphenols found in orange peel extract have demonstrated their effectiveness against mycotoxicosis, positioning orange peel extract as an affordable and renewable agro-waste material for mycotoxin adsorption and toxicity reduction. However, further in vivo studies are needed to confirm the beneficial effects of orange peel extract in livestock and pets affected by mycotoxins.

The different interventions using curcumin, silymarin, grape and olive pomace, and orange peel extracts against mycotoxicosis in livestock, aquaculture, and pet species proposed in this paper have been gathered in Appendix A.

## 3. Conclusions

In conclusion, mycotoxins pose a significant threat to animal and human health, as well as to the global feed supply chain. The chronic effects of mycotoxins on animal health and production, as well as the economic impact on the livestock industry, make it crucial to develop and implement solutions to control and monitor factors implicated in mycotoxicosis. Additionally, natural anti-mycotoxin solutions, such as biological adsorbents, enterosorbents, and natural dietary interventions, are being developed to prevent or mitigate mycotoxicosis and its detrimental effects on animal and human health. Thus, further research on the use of phytogenic feed additives in foodstuffs contaminated by mycotoxins is essential for finding better, cost-effective, environmentally friendly, and generally safe solutions against mycotoxicosis in animal feed. However, the main limitations of using such products lie in the bioaccessibility and bioavailability that can be studied to discover better formulations and conformations of these bioactive compounds. Moreover, the combination of phytogenics with mycotoxin binders is a promising strategy that can enhance their combined effects against mycotoxins. Even though, in this review, we focused on five promising phytogenics with demonstrated positive effects on animal performance and health to present them as potential anti-mycotoxin solutions, many more bioactive compounds yet to be studied could be useful as treatments against the effects of mycotoxins. Finally, most of the studies on anti-mycotoxin phytogenics in the animal industry are focused on livestock and aquaculture species, setting aside companion animals, whose number is growing exponentially worldwide and who are also threatened by the negative effects of mycotoxins, which are especially prevalent in cats and dogs.

## Figures and Tables

**Figure 2 toxins-16-00434-f002:**
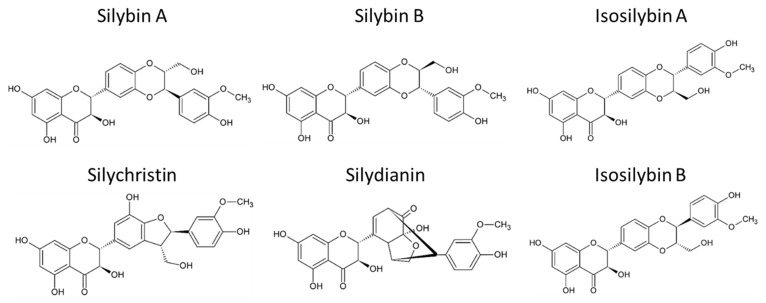
Chemical structure of main flavonolignans contained in silymarin complex.

**Figure 3 toxins-16-00434-f003:**
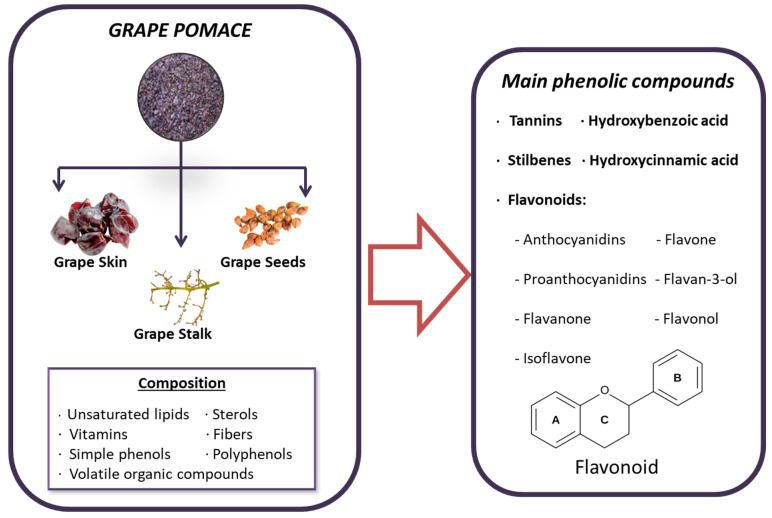
Composition and principal bioactive components of grape pomace extract.

**Figure 4 toxins-16-00434-f004:**
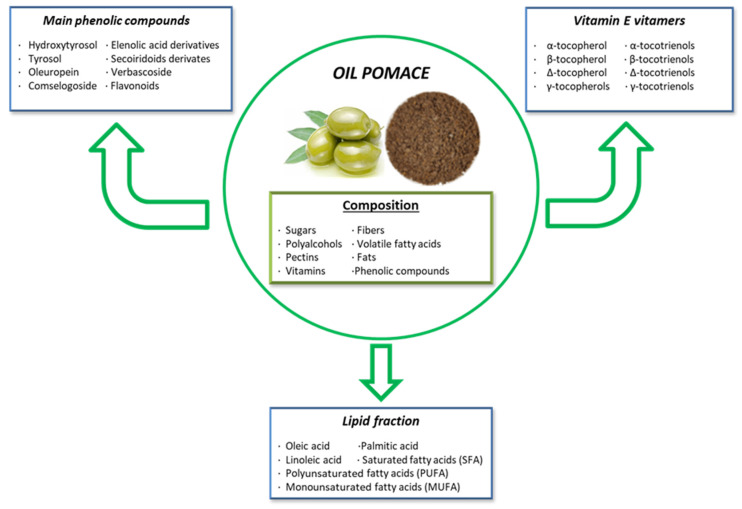
Composition and principal bioactive components of olive pomace extract.

**Figure 5 toxins-16-00434-f005:**
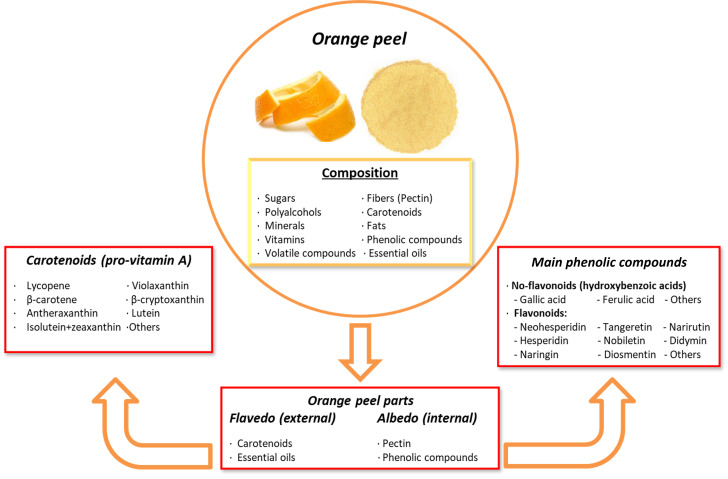
Composition and principal bioactive components of orange peel extract.

## Data Availability

No new data were created or analyzed in this study. Data sharing is not applicable to this article.

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
