# Peer review of "Promising Phytogenic Feed Additives Used as Anti-Mycotoxin Solutions in Animal Nutrition"

_toxins, 2024, doi:10.3390/toxins16100434_

Round 1

Reviewer 1 Report

Comments and Suggestions for Authors

Reviewer: The manuscript summarized 5 different promising phytogenic feed additives that can be used as anti-mycotoxin solutions in animal nutrition. This is a good study on an important topic. However, this review is very descriptive, and I think a more thorough investigation is needed to summarize the current knowledge in this field. I would suggest major revision for this manuscript. 

Major concerns:

1.     Please specify the animal species when talking about the effect of mycotoxin on cellular function or animal growth performance. For example, Line 95-96. AFs are described as genotoxic, hepatotoxic, mutagenic, and carcinogenic and negatively affect the reproduction and the immune system in animals. Please be specific about animal species. Pigs are considered as the most DON-sensitive animal species, followed by mouse > poultry > ruminants.

2.     Why you only chose these 5 phytogenic feed additives in this review? Do you think these 5 feed additives exert their beneficial effect through a common signal pathway? SW Kim lab at NSCU, Ajuwon lab at Purdue and Wang lab at Huazhong Agricultural University, recently presented numerous newly discovered feed additives like, PPARg agonist, quercetin, resveratrol, sodium butyrate etc.

3.     I would suggest summarizing different phytogenic feed additives through a common signaling pathway that can be used to mitigate mycotoxin in animal nutrition, like MAPK pathway, NRF2, PPARg, etc. 

4.     Please make more tables or figures to summarize the current data. It is very to learn something new from current form. 

                   5. Grammar should be checked by a native speaker. 

Comments on the Quality of English Language

                   Grammar should be checked by a native speaker. 

Author Response

REVIEWER 1

Comments and Suggestions for Authors

Reviewer: The manuscript summarized 5 different promising phytogenic feed additives that can be used as anti-mycotoxin solutions in animal nutrition. This is a good study on an important topic. However, this review is very descriptive, and I think a more thorough investigation is needed to summarize the current knowledge in this field. I would suggest major revision for this manuscript. 

Response: Thank you for your thorough review of the manuscript and for helping us improve it. I greatly appreciate the suggestions you provided, and I have addressed all the major concerns you highlighted. Additionally, I carefully revised the entire manuscript to enhance sentence structure, readability, and clarity. The English language was also meticulously reviewed and refined throughout.

Major concerns:

Comment 1:  Please specify the animal species when talking about the effect of mycotoxin on cellular function or animal growth performance. For example, Line 95-96. AFs are described as genotoxic, hepatotoxic, mutagenic, and carcinogenic and negatively affect the reproduction and the immune system in animals. Please be specific about animal species. Pigs are considered as the most DON-sensitive animal species, followed by mouse > poultry > ruminants.

Response 1: I highly appreciate your comment and I agree with the correction about specifying animal species when we talk about the effect of mycotoxin on cellular function or animal growth performance. The following lines will explain the changes we made taking into account your suggestions:

Lines 91-142: I rewrote all the paragraphs: “Firstly, AFs, produced by the genera Aspergillus, are mycotoxins characterized by rapid absorption and metabolization into their active or detoxified metabolites through the liver [3]. These mycotoxins are known for their genotoxic, hepatotoxic, mutagenic, and carcinogenic effects, impacting reproduction and immune systems in both humans and animals, including rodents, poultry, swine, and ruminants. This results in liver damage, decreased productivity, poor eggshell and carcass quality, and increased susceptibility to various diseases. [3,13–15]. Notably, aflatoxin B1 (AFB1) is widely known as the most toxic AF linked to liver damage and diseases in both animals (rodents, livestock, and aquaculture species) and humans, including hepatocellular carcinoma [16,17]. Some species are described as the most susceptible to AFB1, such as turkeys, rats, pigs, sheep, and dogs, whereas others such as monkeys, mice, and chickens are considered resistant [18].

OTs, also produced by Aspergillus and Penicillium species, primarily affect the kidneys, causing nephrotoxicity, kidney enlargement, and tumors in the urinary tract in animals (pigs, poultry, and rodents) and humans [19,20]. They can also disrupt the immune system and fetal development [13]. Ochratoxin A (OTA) is described to be dangerous due to the carry-over of this mycotoxin into animal products from swine, poultry, and even ruminants. The presence of OTA in milk from ruminant species represents a concern for human health [19].

DON, a trichothecene mycotoxin produced by Fusarium species, can reduce feed consumption and weight gain due to symptoms like vomiting, diarrhea, and nausea. DON also causes ribotoxic stress and immune suppression. Swine are the most susceptible species to the toxicity of DON among all the livestock species [19]. On the other hand, contamination by T-2 and HT-2 toxins, which are also trichothecene mycotoxins, can lead to oxidative stress by increased lipid peroxidation, and can mainly trigger carcinogenesis, immune depression, neurotransmitter imbalances, hepatotoxicity, nephrotoxicity, weight loss, growth retardation, oral lesions and reproductive disturbances in humans, poultry and rodents [20,21]. As observed in DON, swine are among the most susceptible animals towards the effects of T-2 toxin [22].

Besides, FBs, also from Fusarium species, are commonly described as sphingolipid metabolism disruptors by inhibiting ceramide synthase and acylation of sphinganine and sphingosine, a biomarker positively correlated to hepatotoxicity, nephrotoxicity, and carcinogenicity [23]. Additionally, FBs can restrict the mitochondrial electron transport chain complex I and promote the formation of reactive oxygen species (ROS), lipid peroxidation, and oxidative stress [24]. FBs affect numerous species, such as horses, pigs, sheep, cattle, fish, poultry, nonhuman primates, mink, rabbits, and rodents. However, natural disease outbreaks primarily occur in horses and pigs, with horses showing greater susceptibility [25].

ZEN, which is also produced by Fusarium species, is characterized to be an estrogen-receptor agonist that can trigger negative consequences, such as alterations in the reproductive tract, fertility impairment, hypoestrogenic syndromes, reduced testosterone and spermatogenesis, and finally cancer in reproductive organs. Exposure to ZEN leads to signs of hyperestrogenism, most notably in pigs, though it can also occur in other animal species to a lesser degree [26,27].

Finally, the emerging and concerning Alternaria mycotoxins, of which the most studied are TeA, AME, and AOH, were described as genotoxic, cytotoxic, carcinogenic, and mutagenic toxins correlated to carcinoma colon cells in humans and esophageal cancer in mammalian cells. However, further research is necessary to truly understand the toxic potential and effects that Alternaria toxins can cause on animal and human health, especially since the European Food Safety Authority (EFSA) advice in 2012 about the concern of high levels of these toxins found in many food products in Europe [28]. Overall, these mycotoxins significantly impact animal health and production, highlighting the need for attention in the livestock industry.”

Comment 2:  Why did you only choose these 5 phytogenic feed additives in this review? Do you think these 5 feed additives exert their beneficial effect through a common signal pathway? SW Kim lab at NSCU, Ajuwon lab at Purdue and Wang lab at Huazhong Agricultural University, recently presented numerous newly discovered feed additives like, PPARg agonist, quercetin, resveratrol, sodium butyrate etc.

Response 2: Thank you for pointing this out. We selected these five phytogenic feed additives—curcumin, silymarin, grape pomace, olive pomace, and orange peel extracts—based on their well-documented anti-mycotoxin properties and demonstrated positive effects on animal health and performance. Each of these compounds has been extensively studied for its bioactive components, which contribute to their antioxidant, anti-inflammatory, and antifungal activities. While these compounds may share some common signaling pathways, such as antioxidant mechanisms or anti-inflammatory responses, their exact modes of action can vary depending on the compound and its bioactive constituents.

Regarding the recent discoveries by SW Kim, Ajuwon, and Wang labs, the additives like PPARγ agonists, quercetin, resveratrol, and sodium butyrate represent exciting areas of research with distinct mechanisms, including modulation of metabolic pathways and immune responses. Our focus in this review was on well-established phytogenics already used as feed additives in animal and human diets. However, these newer additives you mentioned certainly present promising alternatives and could be explored in future studies for their potential synergistic effects with the selected compounds in this review. To highlight this statement, we added this sentence between the lines 260-266:

“These natural compounds not only neutralize mycotoxins but also offer additional health benefits. This section focuses on well-established phytogenics already incorporated into animal and human diets, sharing common signaling pathways like antioxidant and anti-inflammatory mechanisms that boost their antioxidant, anti-inflammatory, and antifungal activities. Additionally, this section explores these phytogenic anti-mycotoxin solutions and their potential impact on animal health and productivity, offering a comprehensive overview of their role in modern animal husbandry.”

Comment 3:   I would suggest summarizing different phytogenic feed additives through a common signaling pathway that can be used to mitigate mycotoxin in animal nutrition, like MAPK pathway, NRF2, PPARg, etc. 

Response 3: Thank you for your suggestion to summarize the different phytogenic feed additives using a common signaling pathway, such as the MAPK, NRF2, or PPARγ pathways, to explain their role in mitigating mycotoxins in animal nutrition. However, after careful consideration, I believe it is more appropriate to maintain the current structure of the manuscript for the following reasons:

Each of the bioactive compounds discussed—curcumin, silymarin, olive pomace, grape pomace, and orange peel—has been extensively studied for its distinct bioactive components, which contribute to its antioxidant, anti-inflammatory, and antifungal properties. While it is true that some of these compounds can activate common signaling pathways, such as the MAPK or NRF2 pathways, their specific mechanisms of action can differ significantly due to the unique composition of bioactive molecules within each compound.

For instance, curcumin is well-known for its strong ability to modulate inflammatory responses through the inhibition of NF-κB and the activation of the NRF2 pathway, but it also has unique effects on apoptosis and cell cycle regulation that are specific to its structure. Silymarin, on the other hand, is characterized by its hepatoprotective properties and its role in inhibiting oxidative stress, particularly in the liver, through the enhancement of glutathione levels and inactivating the CYP450 enzyme.

Similarly, olive pomace and grape pomace, although both rich in polyphenols, exhibit different biological effects due to variations in their phenolic profiles. Olive pomace, with its high content of hydroxytyrosol and oleuropein, primarily exerts antioxidant and anti-inflammatory effects by scavenging free radicals and modulating the MAPK pathway. Grape pomace, which is abundant in resveratrol and proanthocyanidins, has been shown to interact with a broader range of cellular targets, including restore the expression of genes such as eNOS, SOD2, and HO-1, which plays a role involved in reducing hypertension and providing vascular protection

Lastly, orange peel, rich in flavonoids like hesperidin and naringin, shows different bioactive effects, including anti-inflammatory and anticancer properties, through the modulation of oxidative stress markers and downregulation of pro-inflammatory cytokines via the MAPK and NRF2 pathways. The presence of carotenoids and pectin in orange peel also introduces additional bioactive mechanisms, such as lipid peroxidation prevention and liver and gut health enhancement by activating the OXPHOS signaling pathway and enhancing mitochondrial function, and inhibiting hepatic CYP450.

Although these compounds may indeed share common signaling pathways, such as MAPK, NRF2, or PPARγ, their bioactive components influence these pathways in different ways, which is a key factor in their diverse biological effects. Summarizing them under a single signaling pathway could oversimplify the complexity of their actions and obscure the unique therapeutic potential that each compound offers. By maintaining the current structure of the manuscript, we are able to provide a more detailed and nuanced understanding of how these phytogenic compounds contribute to mitigating mycotoxins through their distinct bioactive profiles and mechanisms of action.

Therefore, I believe that retaining the detailed discussion of each bioactive compound is crucial for conveying the full scope of their efficacy and potential applications in animal nutrition. This approach highlights the individuality of each compound, ensuring that their unique contributions are properly represented.

Comment 4. Please make more tables or figures to summarize the current data. It is very to learn something new from the current form. 

I highly appreciate your suggestion. I created a supplementary table summarizing the studies with the phytogenic interventions explained in the manuscript targeting different livestock, aquaculture, and pet species challenged by mycotoxins. I hope it will help to learn something new from the current form. 

I highly appreciate your suggestion. I created a supplementary table summarizing the studies with the phytogenic interventions explained in the manuscript targeting different livestock, aquaculture, and pet species challenged by mycotoxins. I hope it will help to learn something new from the current form. In the manuscript, it was added this sentence to indicate that the studies were collected in a supplementary table: “The different interventions using curcumin, silymarin, grape and olive pomace, and orange peel extracts against mycotoxicosis in livestock, aquaculture, and pet species proposed in this manuscript were gathered in Supplementary Table 1.”

Comment 5. Grammar should be checked by a native speaker. 

The English language was also meticulously reviewed and refined throughout the manuscript.

Reviewer 2 Report

Comments and Suggestions for Authors

The manuscript titled "Promising phytogenic feed additives used as anti-mycotoxin solutions in animal nutrition" reviewed information on phytogenic feed additives and their potential to reduce the toxin effect of mycotoxin in feed on animals and their potential use to mitigate the negative health effects.

The authors were thorough in literature search and compiling the information, however, the authors fells short in presenting a compelling review on the topic.

The authors used words which doesn't fit the context, extra long, sentence, repetition of various information, and lack of logical flow of information.

Because there were numerous locations where the above mentioned issues were present, not all have been captured in the below comments.

The entire manuscript needs thorough revision with correct English sentence formation.

Line 9 – negative economic impact instead of economic losses impact

Line 16 – delete ‘inducing’

Line 17: alleviate instead of ‘eradicate’.

Line 31: delete ‘commodities’

Line 32: long-term existence – revise to suggest persistent in cereal grains.

Line 36: ‘effects’ instead of ‘consequences’

Lines 46-47: around European countries implies countries  surrounding EU countries. Revise to mean countries in EU and NA.

Lines 47-51: organization implicated?? This sentence need rewriting.

Line 46-51: this is very long sentence and the meaning of it unclear. Rewrite into small sentence that are clear.

Line 53-55: Sentence is unclear and need revision.

Line 64: ‘decisive’ meaning

Line 82: Subheading: ‘feed’ instead of ‘nutrition’.

Lines 93-130: This paragraph describes toxic potential and toxic effects of various mycotoxin. While all the relevant information has been included, it was difficult to follow because of use of long sentences, poor organization of sentences, and use of incorrect words based on the context (some examples of these words were included in previous comments). All the above make this paragraph hard to understand. Need rewriting.

Line 131: 60-80% crops – cereal crops?

133-137: Repetition. Most of this text is already included in previous paragraphs.

Comments on the Quality of English Language

The manuscript need a thorough revision of English and sentence formation to increase the readability and comprehension. 

Author Response

REVIEWER 2

Comments and Suggestions for Authors

The manuscript titled "Promising phytogenic feed additives used as anti-mycotoxin solutions in animal nutrition" reviewed information on phytogenic feed additives and their potential to reduce the toxin effect of mycotoxin in feed on animals and their potential use to mitigate the negative health effects.

The authors were thorough in literature search and compiling the information, however, the authors fell short in presenting a compelling review on the topic.

The authors used words which doesn't fit the context, extra-long, sentences, repetition of various information, and lack of logical flow of information.

Because there were numerous locations where the above-mentioned issues were present, not all have been captured in the below comments.

The entire manuscript needs thorough revision with correct English sentence formation.

Response: Thank you for your comprehensive review of the manuscript and help us improve it. I highly appreciate the suggestions you have mentioned above and I modified every point you highlighted to be modified. Moreover, the entire manuscript was thoroughly revised to improve the sentence formation and make it more readable and comprehensive, and the English language was also exhaustively revised.

Line 9 – negative economic impact instead of economic losses impact

Line 9 – I changed ’economic losses impact’ to ‘negative economic impact‘

Line 16 – delete ‘inducing’

Line 16 – I deleted ‘inducing’

Line 17: alleviate instead of ‘eradicate’.

Line 17 – I changed ‘eradicate’ to ‘alleviate‘

Line 31: delete ‘commodities’

Line 31 – I deleted ‘commodities’

Line 32: long-term existence – revise to suggest persistent in cereal grains.

Line 30-32 – I wrote “This stability allows them to persist in cereal grains throughout harvesting, transportation, and storage, leading to long-term contamination” instead of long-term existence

Line 36: ‘effects’ instead of ‘consequences’

Line 32-36 – I changed the sentence to “ As a result, mycotoxins can be transferred from feed to livestock and even into dairy products, posing a significant risk to the global feed supply chain. This affects not only feed safety but also animal health and productivity, ultimately impacting human health through animal-derived products, economies, and international trade”

Lines 46-47: around European countries implies countries  surrounding EU countries. Revise to mean countries in EU and NA.

Lines 47-51: organization implicated?? This sentence need rewriting.

Line 46-51: this is very long sentence and the meaning of it unclear. Rewrite into small sentence that are clear.

Lines 47-50 – I rewrote the sentence: “They are strictly regulated in parts of the European Union and North America due to their impact on livestock feed contamination. Regulatory bodies overseeing food and agriculture have set maximum permissible limits for each type of mycotoxin in animal feed and food products to prevent mycotoxicosis in livestock”

Line 53-55: Sentence is unclear and need revision.

Line 51-54 – In recent years, however, attention has shifted to emerging mycotoxins such as those produced by Alternaria species, particularly alternariol (AOH), alternariol monomethyl ether (AME), and tenuazonic acid (TeA). These toxins have become potential contaminants in livestock feed, prompting increased interest in their biomonitoring and regulation”

Line 64: ‘decisive’ meaning

Line 61: I changed ‘decisive’ to ‘crucial‘

Line 82: Subheading: ‘feed’ instead of ‘nutrition’.

Line 82: I changed ‘nutrition‘ to ‘feed‘

Lines 93-130: This paragraph describes toxic potential and toxic effects of various mycotoxin. While all the relevant information has been included, it was difficult to follow because of use of long sentences, poor organization of sentences, and use of incorrect words based on the context (some examples of these words were included in previous comments). All the above make this paragraph hard to understand. Need rewriting.

Lines 91-142: I rewrote all the paragraphs: “Firstly, AFs, produced by the genera Aspergillus, are mycotoxins characterized by rapid absorption and metabolization into their active or detoxified metabolites through the liver [3]. These mycotoxins are known for their genotoxic, hepatotoxic, mutagenic, and carcinogenic effects, impacting reproduction and immune systems in both humans and animals, including rodents, poultry, swine, and ruminants. This results in liver damage, decreased productivity, poor eggshell and carcass quality, and increased susceptibility to various diseases. [3,13–15]. Notably, aflatoxin B1 (AFB1) is widely known as the most toxic AF linked to liver damage and diseases in both animals (rodents, livestock, and aquaculture species) and humans, including hepatocellular carcinoma [16,17]. Some species are described as the most susceptible to AFB1, such as turkeys, rats, pigs, sheep, and dogs, whereas others such as monkeys, mice, and chickens are considered resistant [18].

OTs, also produced by Aspergillus and Penicillium species, primarily affect the kidneys, causing nephrotoxicity, kidney enlargement, and tumors in the urinary tract in animals (pigs, poultry, and rodents) and humans [19,20]. They can also disrupt the immune system and fetal development [13]. Ochratoxin A (OTA) is described to be dangerous due to the carry-over of this mycotoxin into animal products from swine, poultry, and even ruminants. The presence of OTA in milk from ruminant species represents a concern for human health [19].

DON, a trichothecene mycotoxin produced by Fusarium species, can reduce feed consumption and weight gain due to symptoms like vomiting, diarrhea, and nausea. DON also causes ribotoxic stress and immune suppression. Swine are the most susceptible species to the toxicity of DON among all the livestock species [19]. On the other hand, contamination by T-2 and HT-2 toxins, which are also trichothecene mycotoxins, can lead to oxidative stress by increased lipid peroxidation, and can mainly trigger carcinogenesis, immune depression, neurotransmitter imbalances, hepatotoxicity, nephrotoxicity, weight loss, growth retardation, oral lesions and reproductive disturbances in humans, poultry and rodents [20,21]. As observed in DON, swine are among the most susceptible animals towards the effects of T-2 toxin [22].

Besides, FBs, also from Fusarium species, are commonly described as sphingolipid metabolism disruptors by inhibiting ceramide synthase and acylation of sphinganine and sphingosine, a biomarker positively correlated to hepatotoxicity, nephrotoxicity, and carcinogenicity [23]. Additionally, FBs can restrict the mitochondrial electron transport chain complex I and promote the formation of reactive oxygen species (ROS), lipid peroxidation, and oxidative stress [24]. FBs affect numerous species, such as horses, pigs, sheep, cattle, fish, poultry, nonhuman primates, mink, rabbits, and rodents. However, natural disease outbreaks primarily occur in horses and pigs, with horses showing greater susceptibility [25].

ZEN, which is also produced by Fusarium species, is characterized to be an estro-gen-receptor agonist that can trigger negative consequences, such as alterations in the reproductive tract, fertility impairment, hypoestrogenic syndromes, reduced testosterone and spermatogenesis, and finally cancer in reproductive organs. Exposure to ZEN leads to signs of hyperestrogenism, most notably in pigs, though it can also occur in other animal species to a lesser degree [26,27].

Finally, the emerging and concerning Alternaria mycotoxins, of which the most studied are TeA, AME, and AOH, were described as genotoxic, cytotoxic, carcinogenic, and mutagenic toxins correlated to carcinoma colon cells in humans and esophageal cancer in mammalian cells. However, further research is necessary to truly understand the toxic potential and effects that Alternaria toxins can cause on animal and human health, especially since the European Food Safety Authority (EFSA) advice in 2012 about the concern of high levels of these toxins found in many food products in Europe [28]. Overall, these mycotoxins significantly impact animal health and production, highlighting the need for attention in the livestock industry.”

Line 131: 60-80% crops – cereal crops?

Line 143: We rewrote the sentence adding the word ‘feed crops‘: “Food and agricultural organizations estimated that 60–80% of feed crops are infected by mycotoxins, with 20% of the crops exceeding the European Union (EU) legal food safety limits [29].”

133-137: Repetition. Most of this text is already included in previous paragraphs.

145-151: We modified the paragraph eliminating the repetitive statements: “Thus, chronic mycotoxicosis in feed crops entails additional costs, such as increased costs for health and veterinary care, regulatory fees, and investment in research costs focusing on relieving the impact and severity of the mycotoxin problem, which suppose a millionaire economic impact on the livestock industry [32]. Therefore, developing and implementing solutions to control and monitor factors implicated in mycotoxicosis is highly important to alleviate the negative effects in animal production caused by mycotoxicosis.”

Comments on the Quality of English Language

The manuscript need a thorough revision of English and sentence formation to increase the readability and comprehension. 

Response: The entire manuscript was thoroughly revised to improve the sentence formation and make it more readable and comprehensive, and the English language was also exhaustively revised.

Reviewer 3 Report

Comments and Suggestions for Authors The article is a review and focused on plant-based material to bind mycotoxins found in animal feed. There is a need to find novel methods to reduce livestock exposure to mycotoxins so this is a useful review. I especially like the focus on phytogenic biosorbents because clays have been extensively covered in other review articles.  

My only suggestion is that the introduction is rather are wordy and could use some editing to make it more focused. 

Author Response

Comments: The article is a review and focused on plant-based material to bind mycotoxins found in animal feed. There is a need to find novel methods to reduce livestock exposure to mycotoxins so this is a useful review. I especially like the focus on phytogenic biosorbents because clays have been extensively covered in other review articles.  

My only suggestion is that the introduction is rather are wordy and could use some editing to make it more focused.

Response: Thank you for your detailed review of the manuscript and for assisting us in refining it. I sincerely appreciate your suggestions, and I have made changes to address all the key concerns you raised. Hence, the entire manuscript has undergone a comprehensive revision to improve sentence structure, readability, and overall clarity. 

Round 2

Reviewer 1 Report

Comments and Suggestions for Authors

This revised version improved a lot and I have one more comment below.

Please summarize and compare the difference of these 5 pytohenic products you chose in the paper and other newly discovered additives, like PPARg agonist, quercetin, sodium butyrate, resveratrol, EPA and DHA etc (cite some papers here and make comparsion) in the introduction or discussion part.

For example, although some newly discovered feed additives like PPARg agonist, quercetin, resveratrol, sodium butyrate etc. exert their beneficial effects but hey are ......... that is why we only focus on.....in this review.

Author Response

Comment 1: 

This revised version improved a lot and I have one more comment below.

Please summarize and compare the difference of these 5 pytohenic products you chose in the paper and other newly discovered additives, like PPARg agonist, quercetin, sodium butyrate, resveratrol, EPA and DHA etc (cite some papers here and make comparsion) in the introduction or discussion part.

For example, although some newly discovered feed additives like PPARg agonist, quercetin, resveratrol, sodium butyrate etc. exert their beneficial effects but hey are ......... that is why we only focus on.....in this review.

Response 1: Thank you very much for your suggestion. We agree that incorporating this information could offer a more comprehensive perspective and improve the overall clarity and depth of the discussion. Please note that this content is addressed in lines 73-95.

Reviewer 2 Report

Comments and Suggestions for Authors

It seems like there is an issue with the uploaded file. The corrected text along with the previous version text appear side by side so it is very difficult to read/review. It is unclear what was revised and what wasn't. I have provided few example of the issues below. Please review the file and uploaded revised version as per journal guidelines.

Line 17: both eradicate and alleaviate words appear. Delete eradicate.

Line 26: both metabolite molecules and metabolites appear. Delete metabolite molecules.

Line 35: existencecontamination

Line 37:  representposing a major threatsignificant risk to the

Comments on the Quality of English Language

Not able to review

Author Response

Comment 1: It seems like there is an issue with the uploaded file. The corrected text along with the previous version text appear side by side so it is very difficult to read/review. It is unclear what was revised and what wasn't. I have provided few example of the issues below. Please review the file and uploaded revised version as per journal guidelines.

Response 1: Thank you very much for bringing this problem to our attention. We have re-uploaded the new file and have accepted all the revisions that were made in the previous version. Also, all the changes that have been made in this new version are highlighted in the main body (in yellow).

Comment 2: Line 17: both eradicate and alleaviate words appear. Delete eradicate.

Response 2: Corrected as suggested.

Comment 3: Line 26: both metabolite molecules and metabolites appear. Delete metabolite molecules.

Response 3: Corrected as suggested.

Comment 4:  Line 35: existencecontamination

Response 4: Corrected as requested.

Comment 5: Line 37: representposing a major threatsignificant risk to the

Response 5: Corrected as suggested.

Round 3

Reviewer 2 Report

Comments and Suggestions for Authors

The authors addressed all comments from the reviewers. The current version of the manuscript is acceptable for publication.